# Linking Distributed Leadership with Differentiated Instruction in Inclusive Schools: The Mediating Roles of Teacher Leadership and Professional Competence

**DOI:** 10.3390/bs13120990

**Published:** 2023-11-30

**Authors:** Tiantian Wang, Guoxiu Tian

**Affiliations:** College of Teacher Education, Capital Normal University, Beijing 100048, China; b465@cnu.edu.cn

**Keywords:** distributed leadership, teacher leadership, professional competencies, differentiated instruction, Chinese inclusive education contexts

## Abstract

Despite the clear worldwide school inclusion initiative, translating the widely embraced notions of inclusive education into differentiated teaching practice has been recognized as a common difficulty. Based on replies from 780 educators in inclusive schools in Beijing, China, this study explored how distributed leadership contributes to teachers’ use of differentiated teaching, the mediation role of teacher leadership for inclusion, and teachers’ professional competencies of inclusive education. The results suggest that principals’ distributed leadership directly influences teachers’ employment of differentiated instruction. Teacher leadership for inclusion and professional competencies of inclusive education play a serial mediating role in the relationship between distributed principalship and teachers’ use of differentiated instruction. Implications for implementing inclusive practices were further discussed.

## 1. Introduction

Inclusive education, as an important initiative to include learners with special educational needs (SEN) in regular education classrooms, has been widely adopted in many nations for the purpose of providing equitable and high-quality education for all [1,2,3]. In China, the government launched an initiative called “Learning in Regular Classrooms” (LRC) in the 1980s to make public schools accessible to students with SEN. LRC has also been viewed as China’s indigenous practice of inclusive education [4]. Over the last four decades, the LRC model has gradually evolved from an emphasis on enrollment rate to one on educational quality [5]. In light of the requirement for high-quality inclusive education, regular schools are now expected to serve as inclusive schools, and mainstream teachers are expected to behave as inclusive educators [4,6]. These inclusive schools and educators are expected to provide differentiated instruction to cater to students’ increasingly diverse learning needs. How to promote educators’ use of differentiated instruction in inclusive schools has thus drawn an abundance of attention from both scholars and practitioners.

Differentiated instruction (DI) is theorized as both a teaching philosophy that requests educators to intentionally differentiate teaching content, process, and product [7] and praxis of instruction via which teaching is varied and adapted to match learners’ interests, needs, and capabilities [8]. DI based on instructional adaptations has been recognized as “a cornerstone of effective instruction”, and even “the gold standard that educators should strive for” [9]. However, DI entails at least three major stages: forming a belief, preparing a competence, and implementing the resulting effectiveness [10]. This process relies on a range of organizational and individual antecedents that ought to be consistent with the spirit of DI. Leadership has been identified as a pivotal facilitator in such a process as it affects school culture, interpersonal communication, and faculty capacity [1].

Distributed leadership could emerge as a crucial stimulus in motivating teachers’ enhancement of teaching practices, such as more frequent employment of DI, through its impacts on teachers’ agency, professional competencies, and affective states [11,12]. On the one hand, distributed leadership, to some degree, means a more dispersed, fluid, and shared form of leadership, making possible the activation of teacher leadership by conscious empowerment, shared decision, and collective engagement [13,14]. These teacher leaders are more likely to adopt innovative teaching practices such as DI [15]. On the other hand, by building a collaborative culture where teachers can learn and work together, distributed leadership bolsters teachers’ professional competencies and creates a climate that stimulates teachers’ professional learning and pedagogical improvement [11]. These factors are essential for educators to reinforce the utilization of DI in their daily teaching practice.

Hence, in view of the association among the above factors, this research aims to empirically examine the influence of distributed leadership on teachers’ use of differentiated instruction through the mediating effects of teacher leadership and teachers’ professional competencies of inclusive education in China’s inclusive schools. The central research questions of the present investigation are:To what extent is distributed leadership associated with teachers’ use of differentiated instruction?To what extent do TL and teachers’ professional competencies of inclusive education affect teachers’ use of differentiated instruction?

## 2. Literature Review and Conceptual Framework

### 2.1. Differentiated Instruction (DI)

DI refers to “an approach to teaching in which teachers proactively modify curricula, teaching methods, resources, learning activities, and student products to address the diverse needs of individual students and small groups of students to maximize the learning opportunity for each student in a classroom” [16]. The concept of DI is rooted in Vygotsky’s theory of the zone of proximal development, which invites educators to design instructional practices slightly over students’ present cognitive levels [17]. Scholars have stressed the functions of DI both as a vital part of teaching quality and as a path to ensuring educational equity within a broad educational field, especially in inclusive education [18,19]. Westwood even argued that effective inclusion in mainstream curricula for students with SEN relies on adequate DI matching their capabilities and characteristics [20]. DI could be realized in pedagogical content, process, product, and effect [16]. More specifically, DI practices consist of many implementation approaches, i.e., differentiation of classroom tasks or homework, flexible grouping strategy, modification of assessment and accommodations, and provision of individual feedback [17,21]. Yet, educators’ use of DI could not happen inherently and entails definite contextual conditions. Its successful implementation also depends on educators’ attitudes, values, and competencies toward inclusion. As an important strategy for inclusive classroom practice, effective mechanisms encouraging educators’ employment of DI have attracted rising research attention.

### 2.2. Distributed Leadership (DL) and the Employment of DI

The past two decades have witnessed a spurt of research progress in distributed leadership in the field of educational leadership [22]. Leithwood et al. (2020) asserted that “school leadership has a greater influence on schools and students when it is widely distributed” as one of the strong claims of successful school leadership [23]. A few conceptualizations of distributed leadership have been raised in the existing scholarly work. The first of the most representative concepts was from Spillane [24], who contextualized distributed leadership as a dynamic process and an interactive outcome among leaders, followers, and their situations. Notably, leadership activities rather than leaders’ behaviors emerge as the priority here [25]. This implied that all school members, collectively and collaboratively, make efforts to promote school improvement. The second view pinpoints distributed leadership as shared practices of decision-making, which are carried out by stakeholders from various levels in the whole school [26]. An influential definition of distributed leadership was hereby developed, delineating distributed leadership as one kind of capacity of a school to partake multiple school members in the process of shared decision-making [27,28]. In addition, another different theoretical perspective also identified distributed leadership as one principal leadership style, including typical principalship practices involving teachers in decision-making, developing teacher leadership, empowering teachers, and facilitating collective participation [13,29,30]. This concept has been prevalent in some high-power distance societies (e.g., China, Singapore, South Africa, and Brazil), where the leadership distribution is often ruled by the member in the top power position among the organization (i.e., principals) [31]. In the current study, following the third theoretical view, we conceptualized distributed leadership as a measure of teachers’ perception of principals’ leadership practices and further explored its effects on inclusive practitioners’ use of differentiated instruction in mainland China.

Meanwhile, published studies have shown the potential functions of distributed leadership in gearing teachers’ pedagogical improvements and driving schools toward inclusive values [1,27,32]. However, the influence of distributed leadership on inclusive teaching practices seems to be divergent in the literature. On the one hand, while there have been few explorations on the effects of distributed leadership on such inclusive teaching practices as DI, relatively sufficient empirical evidence suggested positive connections between distributed leadership and different school organizational variables that may motivate educators’ pedagogical improvement, including teacher leadership for inclusion, school supportive climate, faculty collaboration, teacher work engagement, and teacher efficacy [14,33,34,35]. On the other hand, in the field of inclusive schooling, Mullick et al. found that distributed leadership for inclusive education elevates the level at which educators become satisfied with their school working settings [36]. Those with higher satisfaction are more inclined to teach inclusively with more frequent use of DI. Nevertheless, Miškolci et al. claimed that distributed leadership might, under certain circumstances, impede the achievement of inclusive pedagogical activities, especially when leadership responsibilities are distributed to those who hold reserved attitudes toward inclusion or lack of competency to teach inclusively [37]. In sum, there is still a lack of empirical evidence investigating the relationship between principal distributed leadership and teachers’ employment of DI in inclusive contexts, especially in China. Therefore, the present study hypothesizes that:

**H1.** *Distributed leadership significantly and positively affected teachers’ use of differentiated instruction*.

### 2.3. Teacher Leadership (TL) and Its Mediating Role between DL and the Employment of DI

Leading inclusion has brought multi-faceted and complex challenges, and it is difficult for a single “great man” (e.g., principal) to deal with these tasks [38]. Teachers are also expected to assume leadership roles and work collectively with principals to deal with such complex tasks as creating effective inclusive schools and providing high-quality learning experiences for SEN students [39,40]. Teacher leadership has increasingly attracted academic attention in recent years [15]. York-Barr and Duke conducted a pioneering literature review and defined teacher leadership as “the process by which teachers, individually or collectively influence their colleagues, principals, and other members of school communities to improve teaching and learning practices to ultimately increase student learning and achievement” [41]. Wenner and Campbell further summarized the academic work on teacher leadership since 2004, proposing that teacher leadership centers on roles beyond teacher leaders’ classrooms, such as assisting others’ professional learning, engaging in decision-making, and striving forward the whole school’s improvement [42]. Although inconsistently conceptualized and dimensioned, most researchers acknowledged that the ultimate objective of teacher leadership is to influence school-wide teaching practice [43]. Bagley and Tang reported that inclusive teacher leaders exercised leadership mainly through professional support provision, particularly by advocating, innovating, facilitating, or administrating [44]. Wang conceptualized teacher leadership for inclusion as comprising three dimensions: improvement of classroom instruction, facilitation of school-wide partnerships, and engagement of school management and decision-making [45]. Drawing on the statements above, the present research treated teacher leadership for inclusion as a key variable delimited by a process by which educators in inclusive schools influence their colleagues and students to improve inclusive classroom practices to ultimately promote all student learning.

Teacher leadership for inclusion may mediate the hypothesized association between distributed principalship and inclusive educators’ use of DI. On the one hand, the scholarly work illuminates the influence of distributed leadership in facilitating teacher leadership development. Harris asserted early on that a few relations and overlaps exist between distributed leadership and teacher leadership, treating teacher leadership as a resulting form of distributing leadership to teachers to a certain degree [46]. In the inclusive schooling background, Wang and Zhang stressed administrative support from principals as a vital source of teacher agency and an impetus for the enhancement of school-wide inclusive practices [47]. Supporting this argument, Poon-McBrayer and Deng also pointed out that one of inclusive school leaders’ main effective leadership attributes is to create favorable conditions to foster teacher leaders for inclusion [48]. On the other hand, scholars also illustrated how teacher leadership operated to further facilitate pedagogical improvement. Teacher leadership regularly emerges as an “organizational path” from principal leadership to improving instructional practices [23]. For instance, Sebastian et al. (2016, 2017) reported that the indirect effects of principal leadership on student achievement progress would be serially mediated by teachers’ leadership behaviors and classroom teaching practices [49,50]. Likewise, Li and Liu, utilizing integrated school leadership frameworks in Chinese school contexts, revealed that principal transformational leadership can leverage teacher leadership to promote teacher efficacy and student learning [51]. However, research on teacher leadership focusing on inclusion and diversity has been rather deficient [42]. Empirical evidence concerning the relationship among distributed principalship, teacher leadership, and teaching practices in inclusive contexts also remains scarce. Hence, it can be hypothesized that:

**H2.** *Teacher leadership for inclusion significantly mediated the effects of distributed leadership on teachers’ use of differentiated instruction*.

### 2.4. Professional Competence (PC) of Inclusive Education and Its Mediating Role between DL and the Employment of DI

There has been an agreement in contemporary academia that educators’ professional competence in inclusive schooling serves as an important determinant of effective inclusive teaching practices [52]. The core professional competencies of inclusive educators have been internationally acknowledged as three pillars, i.e., attitudes, knowledge, and skills [53,54]. On this basis, Blecker and Boakes pointed out that the necessary dispositions, such as open-mindedness, self-reflection, and a commitment to education equity, were just as important as knowledge and skills in inclusive education [55]. In a study in Beijing, China, teacher agency seeking and acquiring support was identified as the fourth pillar for Chinese inclusive educators [54]. Similarly, Deng et al. reported that Chinese inclusive educators’ professional competencies could be underpinned by four internal dimensions: teaching and instruction, communication and cooperation, attitudes and beliefs, and reflection and development [56]. It is noticeable that such an internal structure of inclusive teachers’ professional competencies might be shaped by external factors, such as school leadership and other contextual variables [47,56].

Teachers’ professional competencies in inclusive education might mediate the relationship between distributed leadership and teachers’ use of DI. On the one hand, principals’ distributed leadership has been demonstrated to be a vital facilitator of reinforcing educators’ professional capacities [57,58]. For instance, Davison et al. reported that the distributed leadership model has effectively built professional capacities for instruction within disciplines [58]. Likewise, Amels et al. argued that the more principals involve the faculty in school reforms and decision-making, the stronger these educators’ competencies to implement educational changes [57]. Meanwhile, Dematthews’s qualitative study also showed that a distributed approach to leadership could make it possible to strengthen the capacity building of inclusive educators and ultimately promote greater inclusion [59]. On the other hand, educators’ professional competencies may be an antecedent of their willingness to change instructional practices. Previous studies suggested that school leadership as an impetus contributed to professional capacity building, further motivating the faculty to take the initiative in improving classroom instruction [46,47]. When educators received competence training on recognizing and addressing diversity in the classroom, they successfully implemented a variety of approaches regarding DI [60]. Consequently, in inclusive settings, it is presumed that principals’ distributed leadership would likely lead to the advancement of teachers’ professional competencies of inclusive education and, therefore, better implementation of DI. In this regard, it can be proposed that:

**H3.** *Teachers’ professional competencies of inclusive education significantly mediated the effects of distributed leadership on teachers’ use of differentiated instruction*.

**H4.** *Distributed leadership influenced teachers’ use of differentiated instruction through the sequential mediation of teacher leadership for inclusion and teachers’ professional competencies of inclusive education*.

### 2.5. The Present Study

The present study aims to explore the relationship between distributed principalship and teachers’ use of differentiated instruction, considering teacher leadership for inclusion and teachers’ professional competencies of inclusive education as serial mediators. Based on the literature review, the present study proposes the following conceptual model (see Figure 1). As mentioned earlier, the rationale for this model is that principals’ distributed leadership practices may be promising to cultivate teacher leaders for inclusion, which contribute to developing their professional competencies of inclusive education, promote the use of differentiated instruction, and teach more inclusively.

## 3. Method

### 3.1. Participants and Procedures

After gaining research ethics approval from the institutional research ethics committee, convenience sampling was used to recruit research participants in Beijing, China. As Beijing is the capital of China, she has a special status in politics, economy, and culture. And she is more developed in elementary education than in other districts in China, including inclusive education. In Beijing, almost all districts have achieved integral inclusive education systems, where accessible public schooling for SEN students is being provided by almost every mainstream elementary school. First, with the assistance of the Beijing Special Education Research Center, research invitations were sent to the inclusive school administrators in two districts, Haidian and Chaoyang. By employing convenience sampling and defining the sampling frame as these two districts, we admit that this is a methodological limitation of this study. Second, these staff helped distribute online questionnaires with consent forms and instruction sheets to teachers in their schools. The data were collected entirely online. The survey questionnaire was designed and completed electronically in Chinese on Wenjuanxing (a popular platform in China providing functions similar to Qualtrics). All participants were informed that they could opt to answer the questionnaire voluntarily, and no incentive would be provided. Ultimately, our final sample consisted of 780 teachers (response rate of 84.50%) from 22 elementary inclusive schools in Beijing.

Demographic information, including district, gender, age, years of teaching, and inclusive education training, was collected from the teachers (see Table 1). Among the participants: (a) 81.7% were female and 18.3% were male; (b) 11.0% were ≤25 years old, 30.4% were 26–35 years old, 28.3% were 36–45 years old, 30.3% were >45 years old; (c) 17.5% had taught for 5 years or less, 35.9% had taught for 6–14 years, 46.6% had taught for 15 years or more; (d) 24.0% frequently participated in inclusive education training, 67.6% occasionally participated in it, and 8.4% had not. All of them had experience interacting with SEN students.

### 3.2. Instruments

A questionnaire including four scales (85 items) was used for data collection. The respondents were required to score items on a six-point Likert scale from “strongly disagree” to “strongly agree”.

#### 3.2.1. Distributed Leadership (DL) Instrument

This study adopted Hairon and Goh’s original DL instrument, which included 25 items [13]. The original instrument had four subscales: bounded empowerment (seven items), developing leadership (five items), shared decision (seven items), and collective engagement (six items). After carrying out confirmatory factor analysis (CFA), the present DL instrument held all four subscales. The four-factor structure showed an acceptable model fit (*χ*^2^ = 805.98, *df* = 65, *p* < 0.001, CFI = 0.947, TLI = 0.936, RMSEA = 0.032, SRMR = 0.018). The wording of some items in this scale was modified to fit the inclusive education context. A sample item includes, “Our principal affirms the importance of shared responsibility for decision-making in developing inclusive education”.

#### 3.2.2. Inclusive Education Teacher Leadership (IETL) Scale

The 22-item IETL scale validated by Wang et al. was utilized to examine inclusive educators’ perceptions of their leadership behaviors [14]. This scale was developed and validated in Chinese contexts for measuring inclusive teacher leaders’ behaviors in five dimensions: advocating inclusive values (three items), implementing inclusive teaching practices (seven items), engaging in school-wide decision-making (four items), encouraging multilateral collaboration (four items), and liaising with an external support system (four items). The CFA results indicated an acceptable model fit (*χ*^2^ = 560.52, *df* = 87, *p* < 0.001, CFI = 0.941, TLI = 0.952, RMSEA = 0.060, SRMR = 0.018), suggesting that IETL is valid and reliable. A sample item is, “I advocate the values of inclusive education in my school”.

#### 3.2.3. Teacher Professional Competence of Inclusive Education Scale

Teachers’ professional competence in inclusive education was gauged using the inclusive teacher professional competence scale [54]. This scale was developed and validated in Beijing, China, having been widely used in Chinese inclusive educational investigations [47]. Its original version includes four dimensions: attitudes, knowledge, skills, and agency toward inclusion. Considering the focus of the present study, we included the first three dimensions. The CFA results for the three-factor model suggested an acceptable model fit (*χ*^2^ = 384.72, *df* = 36, *p* < 0.001, CFI = 0.956, TLI = 0.940, RMSEA = 0.076, SRMR = 0.031). A sample item consists of “I understand the psychological and behavioral characteristics of SEN students in my classroom”.

#### 3.2.4. Differentiated Instruction Scale (DIS)

The 12-item DIS developed by Roy et al. was utilized to measure educators’ employment of DI [8]. It contains two dimensions: instructional adaptations (eight items) and academic progress monitoring (four items). In Roy et al.’s study [8], Cronbach’s alpha for the two sub-scales was 0.86 and 0.74, respectively. The CFA results showed that DIS demonstrated a satisfactory model fit (*χ*^2^ = 472.56, *df* = 42, *p* < 0.001, CFI = 0.935, TLI = 0.929, RMSEA = 0.018, SRMR = 0.037). Sample items include, “I use alternative materials to match students’ abilities”.

### 3.3. Data Analysis

Descriptive statistics and correlation were calculated using SPSS before using Mplus to conduct the structural equation modeling (SEM) analysis. Several indices were used to assess the robustness of fit, including the Chi-square statistic (*χ*^2^), the Tucker–Lewis Index (TLI), the comparative fit index (CFI), and the root mean square error of approximation (RMSEA). Generally speaking, CFI > 0.90, TLI > 0.90, and RMSEA and SRMR < 0.08 were employed as the cutoffs to indicate an acceptable data fit [61]. Concerning mediation analysis, the bootstrap approach was employed to identify indirect impacts [62].

## 4. Results

### 4.1. Descriptive Statistics and Correlations

The descriptive statistics and correlations are presented in Table 2. Among the four variables, DI (M  =  5.03, SD  =  1.11) had the highest mean scores, followed by DL (M  =  4.71, SD  =  0.93), PC (M  =  4.76, SD  =  0.99), and TL (M  =  4.68, SD  =  0.97). All factors showed good reliability with acceptable Cronbach’s α values (0.82–0.91). Meanwhile, significant correlations existed among the four variables. DL was positively related to TL (r = 0.69, *p* < 0.01), PC (r = 0.70, *p* < 0.01), and DI (r = 0.64, *p* < 0.01). TL was positively related to PC (r = 0.54, *p* < 0.01) and DI (r = 0.60, *p* < 0.01). PC was positively associated with DI (r = 0.66, *p* < 0.01).

### 4.2. Hypothesized Model Testing

A SEM model was built to explore the relationship between DL, TL, PC, and ITIC, modeling principals’ DL as a predictor, TL for inclusion, PC of inclusive education as mediators, and teachers’ use of DI as the outcome variable. The results demonstrated that this model reached a good data fit: *χ*^2^ = 404.444, df = 60, *p* < 0.001, CFI = 0.946, TLI = 0.931, RMSEA = 0.078, and SRMR = 0.031. As shown in Figure 2, DL had significant influences on TL for inclusion (*β* = 0.710, *p* < 0.001) and teachers’ use of DI (*β* = 0.318, *p* < 0.001). Therefore, H1 was supported. Additionally, the PC of inclusive education positively related to teachers’ use of DI (*β* = 0.452, *p* < 0.001).

Meanwhile, the mediation analysis results are displayed in Table 3. Neither TL nor PC played a mediating role between principals’ DL and educators’ use of DI separately. However, the total indirect effect of TL and PC on the relationship between principals’ DL and educators’ use of DI was significant at (*β* = 0.303, *p* < 0.001, 95%CI = [0.196, 0.454]). Therefore, H2 and H3 were not supported, but H4 was supported.

## 5. Discussion

### 5.1. Direct Effect of Principals’ Distributed Leadership on Teachers’ Use of Differentiated Instruction

Our data suggested that distributed principalship directly predicts teachers’ employment of DI in inclusive school contexts. This implies that educators are more willing to innovatively implement differentiated teaching practices when school management is more horizontal and decentralized [32,63]. Meanwhile, the results added quantitative evidence in inclusive contexts, resonating with the past qualitative conclusions that emphasized the essential role of distributed principalship in improving inclusive practices and promoting school inclusivity [39,64,65]. When principals delegate leadership and share decision-making with the faculty, educators tend to have higher satisfaction with their organizational working conditions [27], which motivates them to participate more in educational change like inclusive schooling and further boost the quality of instructional practices.

In fact, in collectivist social–cultural contexts like China, principal–teacher relationships are marked by high power distance, obedience to authority, and deference to status [31,66]. Chinese educators have acclimatized to complying with directives and work arrangements from their principals, who enjoyed paramount power in the schools [67]. As confirmed in this study, in such a “principal’s responsibility system” or “principal-control”-based context [68], when principals distribute authority or delegate leadership to inclusive educators, they tend to directly consider implementing inclusion as their working obligation and intend to differentiate teaching. Our finding demonstrated that redistributing school leadership could close the power distance between principals and the faculty, yielding positive outcomes in inclusive education.

### 5.2. The Serial Mediation Effect of Teacher Leadership for Inclusion and Professional Competence of Inclusive Education

First, the SEM results showed that distributed principalship significantly promoted teacher leadership for inclusion, affirming the importance of principals in fostering teacher leaders in inclusive contexts [38]. As Struyve et al. claimed, only when school leaders release power do teachers have the legitimacy to exercise their leadership [69]. Distributed principalship seemed to build an awareness of collective responsibility among inclusive educators [32] and create democratic conditions for them to demonstrate leadership skills [14]. Yet, our study did not confirm that teacher leadership for inclusion directly contributed to teachers’ implementation of DI. There are possible explanations for this finding. On the one hand, in China’s schools with top-down educational administration mechanisms, teachers who enjoy the appointed positions granted by principals may exercise more leadership [70]. Therefore, teacher leadership might be more present in engaging in direction-setting or decision-making toward inclusion at the school level rather than directly improving teaching practices in the classroom. On the other hand, while principals delegate power and authority to the faculty, these so-called teacher leaders lack adequate leadership preparation, positive attitudes, and professional expertise in inclusive schooling [37]. They may passively receive the directives from their leaders but not take effective actions to differentiate teaching intentionally.

Second, educators’ professional competencies of inclusive education positively correlated to their employment of DI. This connoted that educators whose competencies for inclusion were well prepared more frequently differentiate pedagogical approaches. The finding completely resonates with previous research indicating that educators’ capacity building could encourage their behaviors to change teaching [60]. Previous studies have demonstrated that teachers’ level of perceived competencies predicted their intentions to teach inclusively [71]. This study advances a great stride forward by illuminating the statistically significant relationship between teachers’ perceived professional competencies and their actual teaching behaviors, i.e., higher implementation of DI.

In addition, our results evidenced that distributed principalship indirectly influenced educators’ employment of DI through one pathway: the serial meditating effect of teacher leadership for the inclusion and professional competencies of inclusive education. This echoes earlier studies illustrating that the distribution of leadership may cultivate teacher leaders [14] and contribute to teacher capacity through the interchange of practical knowledge, teaching techniques, and expertise regarding the required reform initiatives [25], guiding educators to innovate their daily pedagogical practices [63]. The delegation of power and distribution of leadership by principals may allow teachers to be perceived as respected and approved [69]. Accordingly, these teachers are more willing to exercise leadership roles in inclusive practices in return for their leaders’ psychological trust. As Wang and Zhang observed, such agentic leadership could be a powerful motivator for inclusive educators to develop professional capacities and further differentiate teaching strategies in inclusive classrooms [47]. Our findings further indicated the indispensable role of teacher leadership, without which the avenue between distributed principalship and teachers’ employment of DI was not built. Implementing DI could promote learning in students with diverse abilities and needs and the quality of inclusive schooling, but it brings fresh challenges for educators [8]. With the support and empowerment from principals, inclusive educators are more likely to shoulder leadership activities in the school, but it does not necessarily improve their inclusive teaching practices. Rather, to motivate better implementation of inclusive pedagogical practices, educators must proactively develop professional knowledge, skills, and capacities to practice inclusion on their own initiative.

## 6. Implications and Limitations

To the best of our knowledge, there is an absence of research investigating the effects of school leadership on differentiated instructional practices in inclusive settings. The findings of this study expand the knowledge of what kind of leadership practices are crucial for facilitating differentiated teaching practices and provide practical implications for administrators, principals, and teachers to make informed decisions during inclusive change implementation.

First, our study demonstrated that distributed principalship contributes to teacher leadership and the implementation of DI in inclusive schools. These findings emphasized the indispensable role of principals, who have been responsible for leading greater school inclusion [14]. Therefore, Chinese principals are suggested to provide more leadership opportunities to the faculty and encourage teachers to engage in professional learning and instructional activities for employing differentiated teaching. It might be helpful for principals to take positive measures, such as entrusting experienced inclusive educators to the formal status of the involvement of decision-making, giving inclusive educators professional autonomy, and encouraging collaboration and communication between school leaders and the faculty. Second, the results indicated that teacher leadership for inclusion was significant for boosting teachers’ capacities in inclusive education, which motivated the improvement of inclusive teaching practices. Moreover, implementing DI is reinforced only when inclusive teachers authentically translate leadership into active actions to develop their competencies. This informs us to pay more attention to front-line educators’ ideas and voices in the process of school inclusion. Principals are suggested to be more open, democratic, and benevolent [30], supporting the faculty to be authentic change agents and teacher leaders. School leaders must also make educators feel comfortable shouldering leadership duties instead of being helpless or passive to receive top-down directives when they try to jump out of the previous “teaching comfort zone”. Third, in terms of principal preparation, professional learning and training should be provided to principals, helping these school leaders be aware of the benefits of distributed leadership. In addition, leadership systems, including principal and teacher leadership, should be consciously applied to a school system to build the capacity for inclusive education at a local level (e.g., province, city, or municipality).

There are some limitations to the present research. First, this research is a cross-sectional inquiry, which implies that the authors cannot provide evidence for the causal associations among these variables. Hence, future researchers should employ longitudinal survey design or interview and observation methods to provide causal evidence. The second limitation of this research is that our participants were all from Beijing, China. The regional discrepancies in China and the development of inclusive education in different areas vary widely. Therefore, the results presented here should be explained and implemented more cautiously. In addition, the present exploratory quantitative design could scarcely answer the query of “how leadership is enacted”, which has been a critical topic in the academia of school leadership [24]. Future explorations should adopt more qualitative or mixed methods to further analyze the interactions between school leaders, educators, and specific inclusive schooling contexts.

## Figures and Tables

**Figure 1 behavsci-13-00990-f001:**
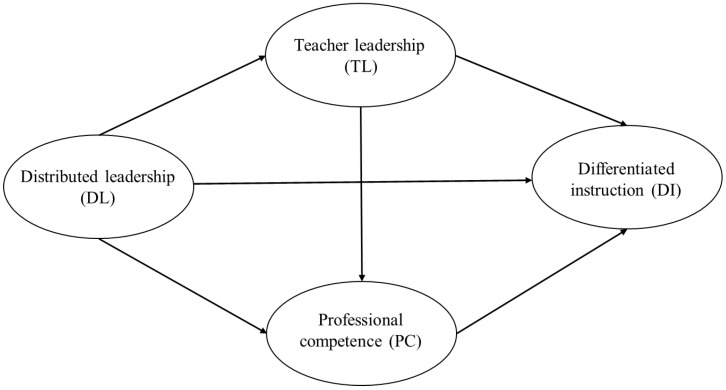
Conceptual model of the study.

**Figure 2 behavsci-13-00990-f002:**
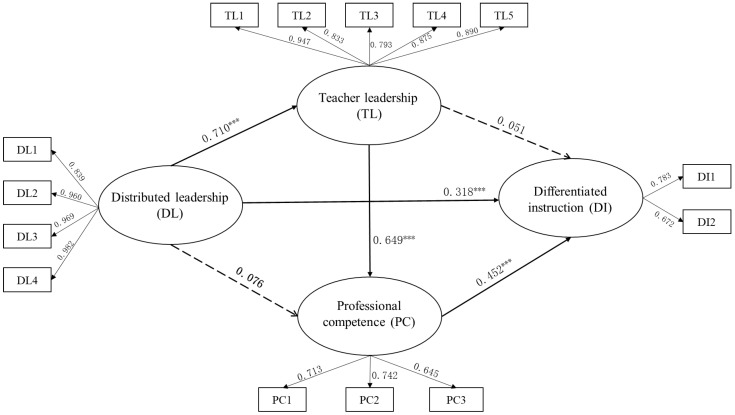
The effects of TL on DI through TL and PC. Note: *** *p* < 0.001; dotted lines indicate non-significant paths.

**Table 1 behavsci-13-00990-t001:** Demographic characteristics of the sample (N = 780).

	N	%
District		
Haidian	432	55.4%
Chaoyang	378	44.6%
Gender		
Female	637	81.7%
Male	143	18.3%
Age		
≤25	86	11.0%
26–35	237	30.4%
36–45	221	28.3%
>45	236	30.3%
Years of teaching		
≤5	137	17.5%
6–15	280	35.9%
>15	163	46.6%
Inclusive education training		
frequently	187	24.0%
occasionally	527	67.6%
never	66	8.4%

**Table 2 behavsci-13-00990-t002:** Descriptive statistics, correlation matrix, and Cronbach’s α.

	M	SD	Cronbach’s α	DL	TL	PC	DI
DL	4.71	0.93	0.82	-			
TL	4.68	0.97	0.83	0.69 **	-		
PC	4.76	0.99	0.91	0.70 **	0.54 **	-	
DI	5.03	1.11	0.88	0.58 **	0.60 **	0.66 **	-

Note: ** *p* < 0.01.

**Table 3 behavsci-13-00990-t003:** Mediation analysis of TL and PC on the effects of DL on DI.

Independent Variable	Via	Mediation Analysis		
		Standard Coefficient	SE	95%CI
DL	TL	0.053	0.092	[0.132, 0.239]
DL	PC	0.050	0.045	[0.033, 0.144]
DL	TL → PC	0.303 ***	0.056	[0.196, 0.454]

Note: *** *p* < 0.001.

## Data Availability

The data presented in this study are available on request from the corresponding author. The data are not publicly available due to confidentiality and research ethics.

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
