# Peer review of "Linking Distributed Leadership with Differentiated Instruction in Inclusive Schools: The Mediating Roles of Teacher Leadership and Professional Competence"

_behavsci, 2023, doi:10.3390/bs13120990_

Round 1

Reviewer 1 Report

Comments and Suggestions for Authors

Dear authors of the manuscript "Linking distributed leadership with differentiated instruction in inclusive schools: The mediating roles of teacher leadership and professional competence",

I hope you are well. I would like to begin by congratulating you on your work and dedication to the research you have done. The study you have conducted on is really interesting and valuable to the academic community. However, after a careful review of your manuscript, I have identified some issues that I believe are important to address in order to improve the quality and clarity of your manuscript.

First, I suggest that you consider including both the overall research objective and the specific hypotheses of the study in the introduction section. Currently they state objective and central research questions, but I think it would be more interesting to add the objective and hypotheses of the study, i.e., what do you hope to achieve after testing what exists in the previous literature? This will help readers better understand what you hope to achieve with your research and what assumptions you are testing.

Organizing the manuscript following the IMR&DC structure (Introduction, Method, Results, Discussion, and Conclusions) would be beneficial. You could integrate your current "Literature Review and Conceptual Framework" section as a subsection of the introduction, and at the end of this section, clearly present the objectives and hypotheses.

In addition, it would be useful to reorganize the subsections in your manuscript so that it is clear which are primary and which are secondary. An example of this is when you set out the instruments used: there is a subsection labeled Instruments and then each of the scales used also appears as a subsection of the general Method section. This can lead to confusion for readers.

Regarding the "Participants and Procedure" section, it would be important to provide the reference approved by your institution's research ethics committee that supports ethical approval of the study. This helps readers to verify that your research meets the necessary ethical standards.

In the Results section, it is essential that you represent your findings in a more detailed and specific manner. When you mention correlations, it would be beneficial to specify the correlations found and not simply state that "correlations were observed between the four variables."

Finally, before submitting your manuscript to the journal, I would recommend that you review and follow the style and formatting guidelines of the journal to which you wish to submit it. Pay special attention to aspects such as references and table layout, as it is important that they conform to the editorial guidelines of the journal.

I hope these comments will be helpful in improving your manuscript and wish you every success in your publication process.

Regards

Author Response

Response to Reviewer 1

Manuscript Number:  behavsci-2655167

Title:   Linking distributed leadership with differentiated instruction in inclusive schools: The mediating roles of teacher leadership and professional competence

Dear reviewer:

Thank you very much for all the very constructive comments. We have taken significant efforts to address step-by-step each of the comments raised and all the main corrections are incorporated in the revised manuscript. Below are our responses (in blue) to all the comments.

Many thanks and best regards,     

all authors

Reviewer #1:

  1. Dear authors of the manuscript “Linking distributed leadership with differentiated instruction in inclusive schools: The mediating roles of teacher leadership and professional competence”, I hope you are well. I would like to begin by congratulating you on your work and dedication to the research you have done. The study you have conducted on is really interesting and valuable to the academic community. However, after a careful review of your manuscript, I have identified some issues that I believe are important to address in order to improve the quality and clarity of your manuscript.

Answer:

Thank you very much for your positive appraisals. The authors wish to express the sincere gratitude to you for taking the precious time and effort to examine the paper.

  1. First, I suggest that you consider including both the overall research objective and the specific hypotheses of the study in the introduction section. Currently they state objective and central research questions, but I think it would be more interesting to add the objective and hypotheses of the study, i.e., what do you hope to achieve after testing what exists in the previous literature? This will help readers better understand what you hope to achieve with your research and what assumptions you are testing.

Answer:

Thank you very much for your constructive comments. According to your suggestions, assumptions have been added in “The present study”. Corresponding changes have also been made in the revised manuscript. As shown below:

Therefore, the present study hypothesizes that:

H1. Distributed leadership had significant and positive effects on teachers’ use of differentiated instruction.

Hence, it can be hypothesized that:

H2. Teacher leadership for inclusion significantly mediated the effects of distributed leadership on teachers’ use of differentiated instruction.

In this regard, it can be proposed that:

H3. Teachers’ professional competences of inclusive education significantly mediated the effects of distributed leadership on teachers’ use of differentiated instruction.

H4. Distributed leadership influenced teachers’ use of differentiated instruction through the sequential mediation of teacher leadership for inclusion and teachers’ professional competences of inclusive education.

  1. Organizing the manuscript following the IMR&DC structure (Introduction, Method, Results, Discussion, and Conclusions) would be beneficial. You could integrate your current "Literature Review and Conceptual Framework" section as a subsection of the introduction, and at the end of this section, clearly present the objectives and hypotheses.

Answer:

Thank you very much for your constructive comments. According to your suggestions, we tried to integrate “Literature Review and Conceptual Framework” as a subsection of the introduction, but we found that if so, the “introduction” section would become so long, and the word number would be more than half of the whole manuscript. Therefore, we chose to reserve the current “Literature Review and Conceptual Framework” section. Anyway, we particularly appreciate your constructive comments.  

  1. In addition, it would be useful to reorganize the subsections in your manuscript so that it is clear which are primary and which are secondary. An example of this is when you set out the instruments used: there is a subsection labeled Instruments and then each of the scales used also appears as a subsection of the general Method section. This can lead to confusion for readers.

Answer:

Thank you very much for your comments. According to your suggestions, the subsections have been reorganized in the revised manuscript, especially in the subsection of “instruments” you mentioned. Meanwhile, the serial numbers have been added to avoid possible confusion for readers.

  1. Regarding the “Participants and Procedure” section, it would be important to provide the reference approved by your institution's research ethics committee that supports ethical approval of the study. This helps readers to verify that your research meets the necessary ethical standards.

Answer:

Thank you very much for your constructive comments. According to your suggestions, “Institutional Review Board Statement” has been attached in the end of the revised manuscript.

This study was conducted in accordance with the Declaration of Helsinki and approved by the Research Ethics Committee of the School of Education of Capital Normal University.

  1. In the Results section, it is essential that you represent your findings in a more detailed and specific manner. When you mention correlations, it would be beneficial to specify the correlations found and not simply state that "correlations were observed between the four variables."

Answer:

Thank you very much for your comments. According to your suggestions, further statements on the correlations have been added in the revised manuscript. As shown below:

Meanwhile, significant correlations existed among the four variables. DL was positively related to TL (r = 0.69, p < 0.01), PC (r = 0.70, p < 0.01) and DI (r = 0.64, p < 0.01). TL was positively related to PC (r = 0.54, p < 0.01) and DI (r = 0.60, p < 0.01). PC was positively associated with DI (r = 0.66, p < 0.01).

  1. Finally, before submitting your manuscript to the journal, I would recommend that you review and follow the style and formatting guidelines of the journal to which you wish to submit it. Pay special attention to aspects such as references and table layout, as it is important that they conform to the editorial guidelines of the journal.

Answer:

Thank you very much for your comments and reminding. According to your suggestions, we revised the references and layout following the guidelines of the journal.

  1. I hope these comments will be helpful in improving your manuscript and wish you every success in your publication process.

Answer:

Thank you very much for your encouragement. Thank you for helping us improve our manuscript.

Reviewer 2 Report

Comments and Suggestions for Authors

Overall, I thought this was a very well written paper. The logic was very clear and reinforced by the literature. As a result, my suggestions are rather minor, and only seek to tighten the arguments discussed. Please see my additional comments throughout the manuscript. 

Intro:

General: I recommend not using the abbreviations in the actual research questions to improve clarity. Also, somewhere, a definition of inclusive education is for sure needed. Yes, this is a term used colloquially, but root it in the literature, and how your study defined it for the purposes of methods, etc. This is critical. 

Framework:

General: There are a lot of abbreviations in this section, which can at times make the content confusing. I don’t have a solution to address this, other than to try not to have so many abbreviations in one sentence. 

Page 2: It’s important to note that DL was originally conceived as something done (or not done) by principals. Right now, that part is missing. Now, we think about DL a little more broadly, but sharing this point to set up your work of TLs is important. 

Page 4: I think a brief discussion of what PC is before going into DL, etc., would be helpful. This just grounds the research approach. 

Methods:

Page 5: 780 teachers out of how many total? Give the population. 

Page 5: because Beijing, China was used, I think a paragraph just about the context is needed here. What makes Beijing unique? What makes it relatable to other contexts in relation to inclusive education? Just be more specific. 

Page 5: I assume this study was part of a larger grant. Please just specify that a little bit more in the first page of the methods, and how this study is unique.

Page 6: I think a table outlining the participant demographics would be a little easier to read. 

Page 6: Was the survey completed electronically? Just be specific.

Page 6: How many questions were actually used? Just be more specific. This is not clear right now.

Page 6: For the DL instrument, which questions were taken of the 25? Just be a little more specific here. 

Results:

General: The results section felt a little short. Though well written, I think a little more detail is required. Despite being a strictly quantitative study, add a little more information. This just helps the reader. Give some context. 

Discussion:

Page 9: I really liked the discussion about China’s high power distance, obedience to authority, and deference to status. I also recommend adding this to your conceptual framing, as these are the cases you are talking about. This just helps bring the paper full circle. 

Implications:

General: I think some mention of how principal preparation in China should emphasize DL would be useful. If China’s context often prohibits DL, than starting with training is absolutely key. This must be discussed in some way. Think of a pipeline to address.

Comments on the Quality of English Language

Few, if any, English issues. Very well written. 

Author Response

Response to Reviewer 2

Manuscript Number:      behavsci-2655167

Title:   Linking distributed leadership with differentiated instruction in inclusive schools: The mediating roles of teacher leadership and professional competence

Dear reviewer:

Thank you very much for all the very constructive comments. We have taken significant efforts to address step-by-step each of the comments raised and all the main corrections are incorporated in the revised manuscript. Below are our responses (in blue) to all the comments.

Many thanks and best regards,     

all authors

Reviewer #2:

  1. Overall, I thought this was a very well written paper. The logic was very clear and reinforced by the literature. As a result, my suggestions are rather minor, and only seek to tighten the arguments discussed. Please see my additional comments throughout the manuscript.

Answer:

Thank you very much for your positive appraisals. The authors wish to express the sincere gratitude to you for taking the precious time and effort to examine the paper.

  1. Intro:

General: I recommend not using the abbreviations in the actual research questions to improve clarity. Also, somewhere, a definition of inclusive education is for sure needed. Yes, this is a term used colloquially, but root it in the literature, and how your study defined it for the purposes of methods, etc. This is critical. 

Answer:

Thank you very much for your constructive comments.

First, according to your suggestion, the abbreviations in research questions have been modified in the revised manuscript.

Second, according to your suggestion, a specific definition of inclusive education in China has been added in the first paragraph of the revised paper. As shown below:

“…… In China, the government launched an initiative called “Learning in Regular Classrooms” (LRC) in the 1980s, to make public schools accessible to students with SEN. LRC has been also viewed as China’s indigenous practice of inclusive education [4]. Over the last four decades, the LRC model has gradually evolved from an emphasis on enrollment rate to one on educational quality [5]. In light of the requirement for high-quality inclusive education, regular schools are now expected to serve as inclusive schools, and mainstream teachers are now expected to behave as inclusive educators [4,6]. These inclusive schools and educators, correspondingly, are expected to provide differentiated instruction to cater for students’ increasingly diverse learning needs. How to promote educators’ use of differentiated instruction in inclusive schools has thus drawn an abundance of attention from both scholars and practitioners.” 

  1. Framework:

General: There are a lot of abbreviations in this section, which can at times make the content confusing. I don’t have a solution to address this, other than to try not to have so many abbreviations in one sentence. 

Answer:

Thank you very much for your comments. According to your suggestions, we tried our best to reduce the use of abbreviations in this section.

  1. Page 2: It’s important to note that DL was originally conceived as something done (or not done) by principals. Right now, that part is missing. Now, we think about DL a little more broadly, but sharing this point to set up your work of TLs is important. 

Answer:

Thank you very much for your constructive comments. We definitely admit that the argument you mentioned is valuable for us to set up our work of TL. We are so sorry that we have not searched for appropriate references to support this argument. Hence, we have not shared this argument in our revised manuscript. We still appreciate your valuable comments. This is a well worth considering direction for our future work.

  1. Page 4: I think a brief discussion of what PC is before going into DL, etc., would be helpful. This just grounds the research approach. 

Answer:

Thank you very much for your constructive comments. In the third paragraph, we mentioned that “Distributed leadership could emerge as a crucial stimulus in motivating teachers’ enhancement of teaching practices, such as more frequent employment of DI, through its impacts on teachers’ agency, professional competences, and affective states”. Therefore, we just briefly mentioned PC and other variables here, and we chose to make a specific discussion for every variable in the section of “Literature review and conceptual framework”.

  1. Methods:

Page 5: 780 teachers out of how many total ? Give the population. 

Answer:

Thank you very much for your comments. In this study, our final sample included 780 of 923 teachers from 22 elementary inclusive schools in Beijing. The response rate was 84.50%. Corresponding changes have also been made in the revised manuscript. As shown below:

“Ultimately, our final sample consisted of 780 teachers (response rate of 84.50%) from 22 elementary inclusive schools in Beijing.”

  1. Page 5: because Beijing, China was used, I think a paragraph just about the context is needed here. What makes Beijing unique? What makes it relatable to other contexts in relation to inclusive education? Just be more specific.

Answer:

Thank you very much for your valuable comments. According to your suggestion, specific description about Beijing have been added in the revised manuscript. As shown below:

“As Beijing is the capital of China, she has a special status in politics, economy, and culture. And she is more developed in elementary education than in other districts in China, including inclusive education. In Beijing, almost all districts have achieved the integral inclusive education systems, where accessible public schooling for SEN students is being provided by almost every mainstream elementary school. First, with the assistance of the Beijing Special Education Research Center, research invitations were sent to the inclusive school administrators in two districts, i.e., Haidian and Chaoyang. By employing convenience sampling and defining the sampling frame as these two districts, we admit that this is a methodological limitation of this study.”

  1. Page 5: I assume this study was part of a larger grant. Please just specify that a little bit more in the first page of the methods, and how this study is unique.

Answer:

Thank you very much for your comments. In fact, this study is an independent study, and it is not a part of a larger grant. According your suggestion, we tried to complement some description to explain the uniqueness of this study. Corresponding changes have been made in “Implication” section. As shown below:

“To the best of our knowledge, there is an absence of research investigating the effects of school leadership on differentiated instructional practices in inclusive settings. The findings of this study expand the knowledge of what kind of leadership practices are crucial for facilitating differentiated teaching practices and provide practical implications for administrators, principals, and teachers to make informed decisions during inclusive change implementation.”

  1. Page 6: I think a table outlining the participant demographics would be a little easier to read.

Answer:

Thank you very much for your comments. According to your suggestion, a table outlining the participant demographics characteristics has been added in the section of “3.1. Participants and procedures”. As shown below:

Table 1

Demographic characteristics of the sample (N=780)

N

%

District

  Haidian

432

55.4%

  Chaoyang

378

44.6%

Gender

  Female

637

81.7%

  Male

143

18.3%

Age

  25

86

11.0%

  26-35

237

30.4%

  36-45

221

28.3%

>45

236

30.3%

Years of teaching

  5

137

17.5%

  6-15

280

35.9%

  ï¼ž15

163

46.6%

Inclusive education training

frequently

187

24.0%

  occasionally

527

67.6%

  never

66

8.4%

  1. Page 6: Was the survey completed electronically? Just be specific.

Answer:

Thank you very much for your comments. This questionnaire survey was completed electronically. According to your suggestion, statements have been added in the revised manuscript. As shown below:

“The data were collected fully online. The survey questionnaire was designed and completed electronically in Chinese on Wenjuanxing (a popular platform in China providing functions similar to Qualtrics). All participants were informed that they can opt to answer the questionnaire voluntarily and no incentive would be provided.”

  1. Page 6: How many questions were actually used? Just be more specific. This is not clear right now.

Answer:

Thank you very much for your comments. A questionnaire including four scales (85 items in total) was utilized for data collection. In the revised manuscript, we complemented the information.

  1. Page 6: For the DL instrument, which questions were taken of the 25? Just be a little more specific here.

Answer:

Thank you very much for your comments. Our participants answered all of 25 questions. Sample item includes “Our principal affirms the importance of shared responsibility for decision making in developing inclusive education”.

  1. Results:

General: The results section felt a little short. Though well written, I think a little more detail is required. Despite being a strictly quantitative study, add a little more information. This just helps the reader. Give some context. 

Answer:

Thank you very much for your comments. We tried to expand the “Results” section. Corresponding changes have also been highlighted in the revised manuscript.

  1. Discussion:

Page 9: I really liked the discussion about China’s high power distance, obedience to authority, and deference to status. I also recommend adding this to your conceptual framing, as these are the cases you are talking about. This just helps bring the paper full circle. 

Answer:

Thank you very much for your constructive comments. Your comments help us improve the coherence of our manuscript. According to your suggestion, some complements have been made. In the “conceptual framework”, there have been some brief description about China’s high power distance. For example, in the section of “2.2. Distributed leadership (DL) and the employment of DI”, we mentioned:

In addition, another different theoretical perspective identified distributed leadership as one principal leadership style, including such typical principalship practices as in-volving teachers in decision-making, developing teacher leadership, empowering teachers, and facilitating collective participation. This concept has been prevalent in some high-power distance societies (e.g., China, Singapore, South Africa, Brazil) where the leadership distribution is often ruled by the member in the top power position among the organization (i.e. principals).

  1. Implications:

General: I think some mention of how principal preparation in China should emphasize DL would be useful. If China’s context often prohibits DL, than starting with training is absolutely key. This must be discussed in some way. Think of a pipeline to address.

Answer:

Thank you very much for your constructive comments. According to your suggestion, the content of “principal preparation” has bee added in the revised Implication section. As shown below:

“Third, in terms of principal preparation, professional learning and training are supposed to be provided to principals, helping these school leaders to be aware of the benefits of distributed leadership.”

Reviewer 3 Report

Comments and Suggestions for Authors

This study investigated the impact of distributed leadership on differentiated instruction, as well as the role of teacher leadership for inclusion and professional competency of inclusive education. Based on responses from educators in inclusive schools in Beijing, it was found that principals' distributed leadership directly affects teachers' use of differentiated instruction. Additionally, teacher leadership for inclusion and professional competency play a mediating role in this relationship. These findings have the potential to greatly benefit several areas within the field of education. For example, they could provide valuable insights for the preparation of administrators in higher education institutes, the development of ongoing leadership support systems in local education agencies, and the creation of policies that promote effective inclusive education practices.

The manscript has been well crafted and logically structured. There are a few minor recommended improvements that can be made to the manuscript before it is published.

page 5, line 225: The acronym ‘BSEC’ should be introduced with the full name of ‘Beijing Special Education Research Center’ in the sentence before.

page 5, line 226: The acronym ‘LRC’ needs to be spelled out.

page 6, line 237: The acronym ‘IE’ needs to be spelled out.

Page 10, line 395 – 416: It would be beneficial to include practical implications, such as its application to a system to build capacity (e.g., through professional learning, coaching, and other systems) of inclusive education or policy changes (e.g., policy-informed practice, practice-enabled policy) at a local level (e.g., province, city, or municipality).

Author Response

Response to Reviewer 3

Manuscript Number:      behavsci-2655167

Title:   Linking distributed leadership with differentiated instruction in inclusive schools: The mediating roles of teacher leadership and professional competence

Dear reviewer:

Thank you very much for all the very constructive comments. We have taken significant efforts to address step-by-step each of the comments raised and all the main corrections are incorporated in the revised manuscript. Below are our responses (in blue) to all the comments.

Many thanks and best regards,     

all authors

Reviewer #3:

  1. This study investigated the impact of distributed leadership on differentiated instruction, as well as the role of teacher leadership for inclusion and professional competency of inclusive education. Based on responses from educators in inclusive schools in Beijing, it was found that principals’ distributed leadership directly affects teachers’ use of differentiated instruction. Additionally, teacher leadership for inclusion and professional competency play a mediating role in this relationship. These findings have the potential to greatly benefit several areas within the field of education. For example, they could provide valuable insights for the preparation of administrators in higher education institutes, the development of ongoing leadership support systems in local education agencies, and the creation of policies that promote effective inclusive education practices.

The manuscript has been well crafted and logically structured. There are a few minor recommended improvements that can be made to the manuscript before it is published.

Answer:

Thank you very much for your positive appraisals. The authors wish to express the sincere gratitude to you for taking the precious time and effort to examine the paper.

  1. page 5, line 225: The acronym ‘BSEC’ should be introduced with the full name of ‘Beijing Special Education Research Center’ in the sentence before.

Answer:

Thank you very much for your comments. According to your suggestions, we added the full name of ‘BSEC’ in the revised manuscript.

  1. page 5, line 226: The acronym ‘LRC’ needs to be spelled out.

Answer:

Thank you very much for your comments. We apologize for the typo in our manuscript

According to your suggestions, we replaced the ‘LRC’ expression into ‘inclusive education’.

  1. page 6, line 237: The acronym ‘IE’ needs to be spelled out.

Answer:

Thank you very much for your suggestion. We have spelled the acronym out in the revised manuscript.

  1. Page 10, line 395 – 416: It would be beneficial to include practical implications, such as its application to a system to build capacity (e.g., through professional learning, coaching, and other systems) of inclusive education or policy changes (e.g., policy-informed practice, practice-enabled policy) at a local level (e.g., province, city, or municipality).

Answer:

Thank you very much for your constructive comments. According to suggestions, we added practical implications in the revised manuscript. As shown below:

“In addition, leadership systems, including principal leadership and teacher leadership, are supposed to be consciously applied to a school system to build the capacity of inclusive education at a local level (e.g., province, city, or municipality).”

Round 2

Reviewer 1 Report

Comments and Suggestions for Authors

Dear Authors:

I would like to express my appreciation for the diligence and effort you have shown in addressing the suggestions and corrections raised in your manuscript "Linking distributed leadership with differentiated instruction in inclusive schools: The mediating roles of teacher leadership and professional competence." 

I have carefully reviewed the modifications made in response to my comments, and I would like to commend you for the way you have addressed each of the above points. Your adjustments have significantly improved the quality and clarity of the manuscript.

I wish you every success in the publication process and look forward to seeing your work published soon.

Best regards. 

Author Response

Thank you very much for your positive feedback.